# Ki67 as a Predictor of Response to PARP Inhibitors in Platinum Sensitive BRCA Wild Type Ovarian Cancer: The MITO 37 Retrospective Study

**DOI:** 10.3390/cancers15041032

**Published:** 2023-02-06

**Authors:** Valentina Tuninetti, Eleonora Ghisoni, Sandro Pignata, Elisa Picardo, Francesco Raspagliesi, Claudia Andreetta, Elena Maldi, Grazia Artioli, Serafina Mammoliti, Lucia Zanchi, Angelica Sikokis, Nicoletta Biglia, Alessandro Parisi, Vincenzo Dario Mandato, Claudia Carella, Gennaro Cormio, Marco Marinaccio, Andrea Puppo, Biagio Paolini, Lucia Borsotti, Giulia Scotto, Margherita Turinetto, Dario Sangiolo, Massimo Di Maio, Giorgio Valabrega

**Affiliations:** 1Department of Oncology, University of Turin, Ordine Mauriziano Hospital, 10128 Turin, Italy; 2Department of Oncology, Immuno-Oncology Service, University Hospital of Lausanne—CHUV, 1011 Lausanne, Switzerland; 3Department of Urology and Gynecology, Istituto Nazionale Tumori, IRCCS-Fondazione G. Pascale Napoli, 80131 Napoli, Italy; 4Obstetrics and Gynaecology 4, Sant’Anna Hospital, AOU Città della Salute e della Scienza of Turin, 88100 Catanzaro, Italy; 5Fondazione IRCCS Istituto Nazionale dei Tumori di Milano, 20133 Milan, Italy; 6Academic Hospital, Azienda Sanitaria Universitaria Friuli Centrale, 33100 Udine, Italy; 7Pathology Unit, Candiolo Cancer Institute, FPO-IRCCS, 10060 Candiolo, Italy; 8UOC di Oncologia Medica, Ulss2 Marca Trevigiana, 31100 Treviso, Italy; 9Oncologia Medica1, IRCCS Ospedale Policlinico San Martino Genova, 16132 Genova, Italy; 10Department of Obstetrics and Gynecology, Fondazione IRCCS Policlinico San Matteo and University of Pavia, 27100 Pavia, Italy; 11Medical Oncology Unit, University Hospital of Parma, 43126 Parma, Italy; 12Obstetrics and Gynaecology Unit, Department of Surgical Sciences, Umberto I Hospital, School of Medicine, University of Turin, 10124 Turin, Italy; 13Department of Life, Health and Environmental Sciences, University of L’Aquila, 67100 L’Aquila, Italy; 14Department of Oncology, Università Politecnica delle Marche, Azienda Ospedaliero-Universitaria Ospedali Riuniti di Ancona, 60126 Ancona, Italy; 15Unit of Obstetrics and Gynaecology, Azienda Unità Sanitaria Locale—IRCCS, 42122 Reggio Emilia, Italy; 16Medical Oncology Unit, IRCCS Istituto Tumori “Giovanni Paolo II”, 70124 Bari, Italy; 17Gynecologic Oncology, IRCCS Istituto Tumori “Giovanni Paolo II”, 70124 Bari, Italy; 18Department of Interdisciplinary Medicine University of Bari, 70124 Bari, Italy; 192nd Unit of Obstetrics and Gynecology, Department of Biomedical and Human Oncological Science (DIMO), University of Bari, 70124 Bari, Italy; 20Gyn-Obst Unit, S. Croce e Carle Hospital, 12100 Cuneo, Italy; 21SC Anatomia Patologica 1, Fondazione IRCCS Istituto Nazionale dei Tumori, 20133 Milan, Italy; 22SC Direzione Sanitaria, Ordine Mauriziano Hospital, 10028 Turin, Italy; 23Department of Oncology, University of Turin, 10124 Turin, Italy

**Keywords:** ovarian cancer, PARP inhibitor, Ki67, niraparib, rucaparib

## Abstract

**Simple Summary:**

Low cost, reliable predictors of benefit from PARP inhibitors (PARPi) are missing for relapsed BRCA wild-type (WT) ovarian cancer (OC). MITO 37 is a multicenter retrospective study aiming at correlating Ki67 expression at diagnosis with a clinical outcome following platinum treatment and PARPi maintenance. Clinical data were collected from 150 patients with high grade serous or endometroid BRCA WT OC treated with niraparib or rucaparib maintenance in 15 centers within MITO group. Ki67 expression was assessed by certified pathologists on tumor tissue at diagnosis and median Ki67 was used as cut-off. 136 patients were included. Median Ki67 was 45.7% (range 1.0–99.9). No statistically significant differences in response to PARPi neither in progression free survival and overall survival were identified between low and high Ki67 subgroups. High Ki-67 at diagnosis cannot discriminate responders to PARPi among OC BRCA WT patients.

**Abstract:**

Background: There is compelling need for novel biomarkers to predict response to PARP inhibitors (PARPi) in BRCA wild-type (WT) ovarian cancer (OC). Methods: MITO 37 is a multicenter retrospective study aiming at correlating Ki67 expression at diagnosis with a clinical outcome following platinum treatment and PARPi maintenance. Clinical data were collected from high grade serous or endometroid BRCAWT OC treated with niraparib or rucaparib maintenance between 2010–2021 in 15 centers. Ki67 expression was assessed locally by certified pathologists on formalin-fixed paraffin embedded (FFPE) tissues. Median Ki67 was used as a cut-off. Results: A total of 136 patients were eligible and included in the analysis. Median Ki67 was 45.7% (range 1.0–99.9). The best response to platinum according to median Ki67 was 26.5% vs. 39.7% complete response (CR), 69.1% vs. 58.8% partial response (PR), 4.4% vs. 1.5% stable disease (SD). The best response to PARPi according to median Ki67 was 19.1% vs. 36.8% CR, 26.5% vs. 26.5% PR, 26.5 vs. 25% SD, 27.9% vs. 16.2% progressive disease (PD). No statistically significant differences in progression free survival (PFS) and overall survival (OS) were identified between low and high Ki67. PFS and OS are in line with registration trials. Conclusions: Ki67 at diagnosis did not discriminate responders to PARPi.

## 1. Introduction

Major clinical developments in the treatment of ovarian cancer (OC) have occurred during the past ten years, including the introduction of angiogenesis inhibitors [1] and poly-ADP ribose polymerase inhibitors (PARPi) [2,3,4,5], which have significantly increased overall survival (OS) rates [6]. Despite these advancements, OC is still the most lethal gynecologic malignancy accounting for more than 19880 new cases and 12810 deaths worldwide [7,8]. Indeed about 70% of OC treated with first-line surgery and carboplatin-paclitaxel combination will eventually relapse within two years after diagnosis [9].

At relapse, the first evaluation to be made is to determine whether a patient is a candidate for systemic anticancer therapy or surgery and is willing to continue treatment. Tumor biology, prior therapies, response to chemotherapy, therapy free interval (TFI), persistent toxicities, patient preference, and symptoms must all be considered when deciding whether to offer platinum-based therapy or non-platinum treatment [9].

Platinum-based chemotherapy followed by a PARPi maintenance could be considered for patients with a platinum-free interval (PFI) ≥ 6 months. Three PARP inhibitors are currently approved for the maintenance treatment of platinum-sensitive OC: niraparib, olaparib, and rucaparib for patients with germline or somatic *BRCA 1/2* mutation, niraparib and rucaparib also for serous *BRCA 1/2* wild-type (WT) patients [10,11,12,13]. The choice of the treatment depends mainly on safety profile since different trials show similar activity of the two drugs. Moreover, in Italy, endometrioid ovarian cancers can only be treated with rucaparib. No data are available on comparison between the two drugs [14,15].

However, based on recent data suggesting a detrimental effect of PARPi on OS in several randomized trials in relapsed BRCA WT OC, including homologous recombination deficiency (HRD) population [16,17,18], the Food and Drug Administration (FDA) have restricted their indication at platinum sensitive relapse to BRCA-mutated patients. Although the exact scenario on the use of PARPi is still to be defined (the European Medicines Agency has maintained the current indication), it is clear that current HRD testing still present several limitations. Different approaches are currently being investigated to identify *BRCA1/2* WT tumours that can benefit from DNA-damaging agents and PARPi based on the presence of HRD, that is, (1) scores capturing large genomic aberrations, so-called ‘genomic scars’, (2) analysis of mutational signatures, or (3) point mutations identified in homologus recombination repair (HRR) genes using DNA sequencing panels.

Commercial HRD tests such as Myriad and Foundation One [19] identify genomic scars and not the actual genomic status neither the mechanisms of resistance that develop during therapy and functional information of the HRD pathway’s activity, thus translating in a weak predictive value of response to PARPi [5,20,21,22,23,24]. Some limitations of these and other available tests also include the proportion of samples returned with inconclusive results, false-negative results, and high cost. This is the reason why the academic community is looking at more “functional tests” in tissues, such as RAD51 foci expression [25].

Therefore, the search for an easily accessible, low-cost, reproducible biomarker represents an urgent need in clinical practice.

In the early 1980s, Scholzer and Gerdes discovered the Ki67 antigen, which encodes two protein isoforms with molecular weights of 345 and 395 kDa [26]. Ki67 protein has a half-life of only 1 to 1.5 h. It is present in all active cell cycle phases (G1, S, G2, and M), but not in resting cells (G0) [27,28]. Ki67 expression is nuclear, and it is proportional to the mitotic count but reveals more proliferating tumor cells than the citoplasmatic expression since it marks cells not in the G0 phase.

Interestingly, although with different cut-offs varying among different studies, high Ki67 is predictive of pathologic complete response (pCR) in breast cancer undergoing neoadiuvant chemotherapy with alkylating agents, anthracyclines, and taxanes [29,30]. In the context of OC, some reports correlated high Ki67 expression to poor histological and clinical characteristics. In particular, Kritpracha et al. [31], showed that in 105 patients with locally advanced OCs, the percentage staining of Ki67 expression ranged from 0.3 to 100%, with a median of 11.9% and that Ki67 was higher in serous tumors than in other histotypes (*p* = 0.048). The 5-year OS was 15.1% in the high Ki67 (≥11.9%) and 36.5% in the low Ki67 (<11.9%) patients, respectively. Median overall survival (OS) in the two groups was 1.8 years and 3.0 years, respectively (*p* < 0.008).

Heeran et al. [32] analyzed Ki67% in 606 OC patients using 10% as the cut-off level and found that the frequency of Ki67% expression increased with an increasing International Federation of Gynecology and Obstetrics (FIGO) stage (*p* = 0.003) and histological grade (*p* > 0.0001).

Finally, Layfield et al. [33] demonstrated that Ki67 had prognostic significance in late-stage OC. The median OS of patients whose carcinoma had a high Ki67 expression (defined as >15%) was 30 months compared to 16 months in the low-expression subgroup.

The aim of our study was to investigate the role of Ki67 as a potential biomarker of response to platinum salts and PARPi in relapsed OC.

## 2. Material and Methods

### 2.1. Patients

MITO 37 is a multicenter retrospective Italian study aiming at correlating Ki67 expression with a clinical outcome following platinum treatment and PARPi maintenance. Clinical data were collected from all patients with high grade serous or endometroid BRCA WT ovarian cancer treated with niraparib or rucaparib maintenance between 2010–2021. The study has been approved by the ethical committee of participating Institutions. We collected data from 15 Italian Centers. See Appendix A.

Eligible patients were at least 18 years of age and had platinum-sensitive (defined as PFI ≥ 6 months) relapsed cancer of the ovary, peritoneum, or fallopian tube (collectively defined as ovarian cancer), with disease progression after at least 1 line of chemotherapy. All patients had high-grade serous or endometrioid tumors that were classified at diagnosis according to the FIGO staging criteria. Before the start of PARPi, all the patients had received four to six cycles of platinum-based chemotherapy, which had resulted in a complete response (CR) or partial response (PR), according to investigator assessment. Patients receiving treatment within clinical trials were not eligible. All patients had been tested for *BRCA1/2* germline and/or somatic mutations and resulted WT.

The primary endpoint was to determine, in a real-world setting, the overall response rate (ORR) to platinum-based chemotherapy and to PARPi according to the Ki67 value expressed as %. Patients were divided into 2 groups (low and high) using the median value as the cut-off (see below).

The secondary endpoints were to describe progression free survival (PFS) and OS comparing subgroups based on the Ki67 value.

PFS was defined as the time from the first day of niraparib or rucaparib administration to the date of objective disease progression on imaging according to the response evaluation criteria in solid tumors (RECIST), version 1.1, or death from any cause. Patients who did not experience progression disease (PD) were censored on the date of the last follow-up visit. OS was defined from the first day of niraparib or rucaparib administration to death for any cause or the last follow up visit. The median follow-up (FU) was calculated according to the reverse Kaplan–Meier formula. PFS and OS were calculated according to the Kaplan–Meier formula, and groups were compared by log-rank test.

Within 8 weeks after completion of the last dose of platinum-based chemotherapy, patients were assigned to receive oral niraparib or rucaparib as per clinical practice choice (according to previous toxicities, comorbidities, and patients’ choice) in 28-day cycles.

All the patients started rucaparib at a fixed dose of 600 mg twice daily, while patients starting niraparib were assigned a full dose of 300 mg daily or an individualized dose of 200 mg daily according to platelets count and weight (a 200 mg daily dose was assigned in case of a baseline body weight of less than 77 kg, a platelet count of less than 150,000/mm^3^, or both) [34].

The data cut-off for the analysis was 30 April 2022.

### 2.2. Ki67 Staining and Scoring

The Ki67 expression was assessed locally by certified pathologists on a formalin-fixed paraffin-embedded tumor tissue at diagnosis.

To overcome the possible bias of poor reproducibility, a kick-off meeting with the pathologists was performed in order to define the criteria for Ki67 evaluation.

Tissue samples from each tumor lesion were fixed for 24 h in 10% buffered formalin. The selected blocks were cut at a 3 mm thickness and immunohistochemistry (IHC) was performed using a Ki67 mouse monoclonal antibody (Mib-1 clone, monoclonal, Dako, Glostrup, Denmark), then incubated with a commercially available detection kit (EnVision FLEX; Dako, Glostrup, Denmark).

Slides were explored with a conventional light microscope by an experienced pathologist trained in Gynecopathology. A representative field for Ki67 evaluation was selected. In case of heterogenous staining, a hot spot (an area with high Ki67 expression) was chosen. Ki67 expression was analyzed for each case, and all colored nuclei were considered positive, independent of the intensity and distribution. It was performed using a quantitative method with evaluation of the ratio of stained cells compared with the total number of cells in the field. A total of 100 cells were evaluated for each case (Figure 1).

## 3. Results

Baseline demographic and clinical characteristics of patients included in the MITO 37 study are summarized in Table 1.

Three patients were excluded because they received PARPi as a first-line maintenance treatment and not at relapse and one patient was excluded because she received olaparib. At diagnosis for 10 patients, a tumor tissue was not available for Ki67 assessment. Therefore, the final efficacy analysis was performed on 136 patients (Figure 2).

The median age at diagnosis was 63 years (range 38–84). All patients were somatic and/or germline BRCA wild-type as per the inclusion criteria. In particular, the germline BRCA 1–2 wild-type status was known in 87.5% and the somatic *BRCA1/2* wild-type status was known in 61.8%.

At diagnosis, the majority of the patients (82.3%) had advanced disease (stage FIGO III-IV), high grade serous ovarian cancer (HGSOC) histology (92.6%), while the remaining patients had OC with endometrioid histology (5.9%) or mixed (serous and endometrioid) histology (1.5%).

Most patients (76.5%) had an elevated CA-125 serum level at diagnosis.

More than half of the patients (66.2%) underwent upfront debulking surgery and had no residual disease (R0, 70.6%).

Initial PFI (time between the last cycle of platinum during first line and evidence of disease progression) was ≥12 months in 75.7% of the cases and between 6 and 12 months in 22.8%.

The CA-125 serum level was normalized before starting PARPi maintenance in the majority (68.4%).

Most of the patients (72.1%) received PARPi (niraparib or rucaparib) at first recurrence, after a second line of platinum-based chemotherapy, while the remaining received PARPi after three or more lines of platinum-based chemotherapy.

Most patients had received bevacizumab as maintenance at first line or in a line before starting PARPi (67.6%).

The two most used platinum-based regimens at relapse were carboplatin-liposomal doxorubicine and carboplatin-paclitaxel, respectively, 34.6% and 33.1%.

About one third (33.1%) of the patients achieved CR with platinum-based chemotherapy and 64% achieved PR, while 2.9% had a stable disease (SD) at imaging after platinum-based chemotherapy, but they were considered eligible to start PARPi in a clinical practice, due to a CA-125 level decrease.

About one third of the patients (33.8%) underwent surgery at relapse (considering any relapse before PARPi).

The median FU was 29.1 months (range 24.7–33.5).

As shown in Table 2, the majority of the patients (90.0%) started niraparib as maintenance after platinum-based chemotherapy; among these, 38 patients (31.4%) started at a full dose of 300 mg daily, while 83 patients (68.6%) started an individualized dose of 200 mg daily according to platelets count and weight.

All patients in the rucaparib group started at a full dose of 600 mg twice daily. About half of the patients reduced their dose of PARPi during maintenance due to toxicities (48.5%), and at the data cut-off, 22.6% of patients were still ongoing with maintenance therapy. The best response to PARPi was CR (30.9%); however, 24.3% experienced PD at first evaluation. The majority of the patients discontinued maintenance therapy due to PD (72.8%).

The median treatment duration was 9.3 months (95%, CI 7.9–10.7 months).

As for patients who experienced disease progression after PARPi, the subsequent chemotherapy line was mainly platinum-based chemotherapy (32.4%) and, at the data cut-off, 61.8% of the patients were alive.

### Ki67 Assessment and Predictive Value

The median Ki67 expression was 45.7% (range 1–99.9).

The patients’ characteristics according to the median Ki67 (low ≤ 45.7% vs. high > 45.7%) are summarized in Table 3 and Table 4.

The two groups (≤45.7% vs. >45.7%) were well balanced in terms of baseline characteristics with the exception of bevacizumab use: patients with high Ki67 had received bevacizumab more frequently than those with low Ki67.

The median PFS in the whole cohort was 11.4 months (95% confidence interval [CI] 8.2–14.6). No difference according to median Ki67 value was detected: for patients with a low Ki67 value, the median PFS was 11.4 months (95%CI 8.2–14.6); for patients with a high Ki67 value, the median PFS was 10.6 months (95%CI 6.4–16.4), HR 1.31 (95% CI 0.89–1.94), *p*-value= 0.18 (Figure 3).

The median OS was 37.1 months (95% CI 37.1–42.6). Also for OS, no statistical significant difference according to the median Ki67 value was observed between the two groups: for patients with a low Ki67 value ≤ 45.7, the median OS was 34.9 months (95% CI 21.5–48.3); for patients with a high Ki67 value > 45.7, the median OS was not reached (Figure 4).

## 4. Discussion

The MITO 37 study investigated the predictive role of Ki67 IHC expression in serous or endometrioid OC patients receiving PARPi (niraparib and rucaparib) at platinum-sensitive recurrence.

To date, no data are available about the potential role of Ki67 IHC expression in predicting PARPi response, but several reports showed that OC patients who have a high proliferation index (PI) are more likely to have poor prognostic factors, including an advanced FIGO stage, a higher tumor grade, a bulk residual tumor, and a poor response to chemotherapy, and also have a less favorable 5-year survival compared with those with a low PI [31,32,33].

In our study, in the absence of a universally accepted cut-off, we used the median value to divide the population into two categories (low vs. high Ki67).

Unfortunately, we were not able to identify significant differences between low and high Ki67 cancers in terms of: age at diagnosis, FIGO stage, grading or histological subtype, CA-125 level at diagnosis, type of surgery at diagnosis (upfront, IDS, and no surgery) and residual disease at first surgery, CA-125 before starting PARPi, BRCA status, first PFI, number of previous lines of chemotherapy, clinical response to platinum, number of surgeries before PARPi, and residual disease at last surgery. Interestingly, and perhaps due to its earlier approval, niraparib was clearly predominant for the treatment of the patients included in the study.

The median PFS in our study was 11.7 months and was consistent with the NOVA and ARIEL-3 studies. In the NOVA trial [15], in fact, the median PFS was 9.3 months and in ARIEL-3 [14], it was 13.7 months. No differences according to the median Ki67 value have been detected also for PFS.

The present study contains both limitations and strengths. Limitations include those associated with the retrospective nature and the intrinsic risk of confounding the relatively small number of patients included.

Ki67 was assessed mostly at diagnosis and not from a tumor tissue at relapse, just before PARPi was started. Considering tumor clones’ evolution during the natural history of the disease, our data might not reflect the Ki67 value at the time of relapse.

Furthermore, such information comes from analyses of single sites of tumor growth. Considering the biology of ovarian cancer and its extremely heterogeneous nature, both spatially and temporally, it is legitimate to state that coexistence of several different clones at diagnosis render unpredictable which ones would be prevalent at relapse [35,36].

In conclusion, our study was not able to demonstrate a potential role for Ki67 expression to predict response to PARPi at diagnosis nor to identify different populations in terms of clinical features.

## 5. Conclusions

This work is an example of how the MITO and other collaborative groups could collaborate efficiently and address important clinical issues such as the identification of reliable biomarkers of response to PARPi for BRCA WT-relapsed ovarian cancer patients. Although the MITO 37 was a negative study, we believe we should increase the recognition among all clinicians that negative studies are also necessary to the advancement of clinical practice. Publication of negative results is particularly useful to avoid publication bias, which could affect not only studies of interventions but also studies exploring the performance of diagnostic, prognostic, and predictive factors.

Moving into perspective, we are planning to centralize Ki67 readings and to enrich the study including an analysis of tumor tissues at relapse and possibly multiple sites.

Finally, this work reminds us of the complex nature of OC and highlights the urgent and still-unmet need for early and accessible in-tissue biomarkers to predict the response to PARPi and avoid unnecessary toxicities to patients.

## Figures and Tables

**Figure 1 cancers-15-01032-f001:**
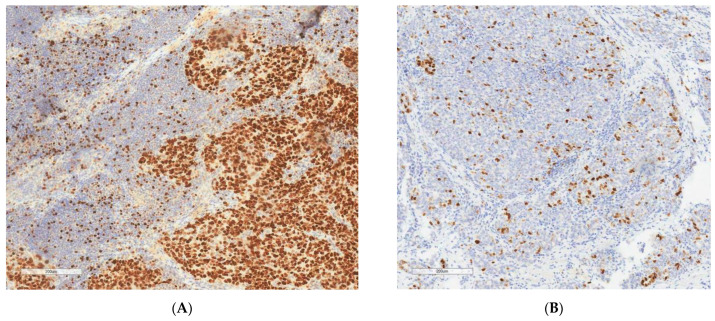
Ki67 IHC staining on ovarian cancer tissue. Original magnification 200×. (**A**) high Ki67 (>45.7%). (**B**) low Ki67 (<45.7%).

**Figure 2 cancers-15-01032-f002:**
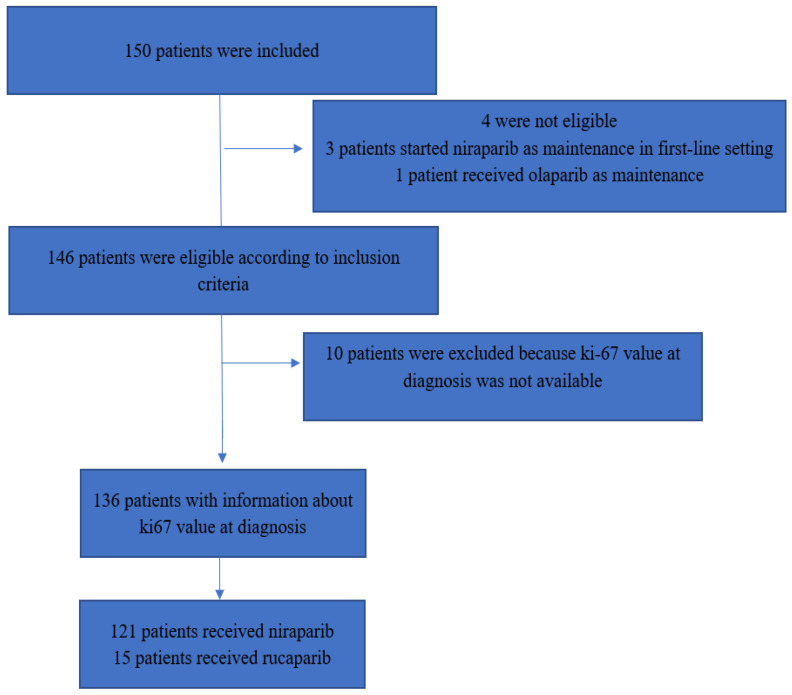
Flowchart of the study.

**Figure 3 cancers-15-01032-f003:**
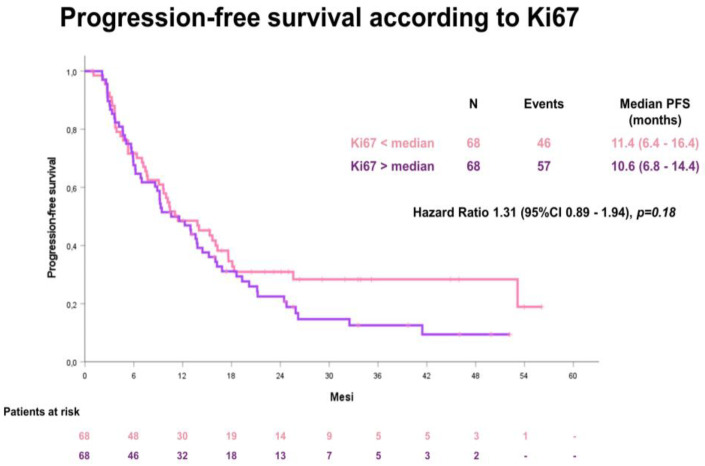
Progression-free survival (PFS) according to the median Ki67 value.

**Figure 4 cancers-15-01032-f004:**
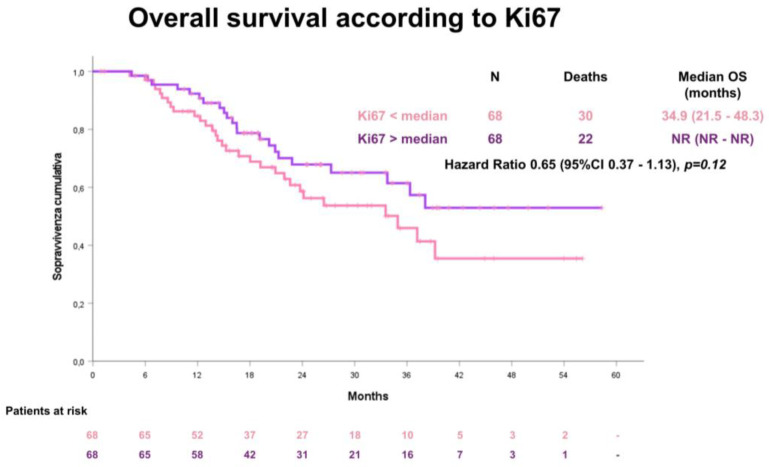
Overall survival (OS) according to the median Ki67 value.

**Table 1 cancers-15-01032-t001:** Demographic and clinical characteristics of the patients at baseline.

Characteristic	N	%
***Median Age at Diagnosis* (Range) **	63 (range 38–84)	
***International FIGO stage at diagnosis*** I II III IV NA	8/136 15/136 91/136 21/136 1/136	5.9% 11% 66.9% 15.4% 0.7%
***Grading*** G2 G3	3/136 133/136	2.2% 97.8%
***CA-125 level at diagnosis*** ≤ULN >ULN NA	15/136 104/136 17/136	11% 76.5% 12.5%
***Type of surgery at diagnosis*** Upfront IDS No surgery	90/136 44/136 2/136	66.2% 32.4% 1.5%
***Residual disease at first surgery*** R = 0 R > 0 NA	96/136 39/136 1/136	70.6% 28.7% 0.7%
***CA-125 before starting PARPi*** ≤ULN >ULN NA	93/136 41/136 2/136	68.4% 30.1% 1.5%
***Histologic type*** Serous Endometrioid Mixed serous and endometrioid	126/136 8/136 2/136	92.6% 5.9% 1.5%
***BRCA wild-type (germline)*** Yes NA	119/136 17/136	87.5% 12.5%
***BRCA wild-type (somatic)*** Yes NA	84/136 52/136	61.8% 38.2%
***First PFI*** ≥12 months 6–12 months NA	103/136 31/136 2/136	75.7% 22.8% 1.5%
** *No. of lines of chemotherapy before PARPi* **		
2 lines (first recurrence)	98/136	72.1%
>2 lines	38/136	27.9%
3 lines	26/38	
4 lines	6/38	
5 lines	5/38	
7 lines	1/38	
***Bevacizumab*** Yes No	92/136 44/136	67.6% 32.4%
***Type of chemotherapy before PARPi*** Carboplatin-liposomial doxorubicine Carboplatin-gemcitabine Carboplatin monotherapy Carboplatin-paclitaxel Other	47/136 23/136 1/136 45/136 20/136	34.6% 16.9% 0.7% 33.1% 14.7%
***No. of surgeries before PARPi*** 1 2 3 5	90/136 40/136 5/136 1/136	66.1% 29.4% 3.7% 0.7%
***Residual disease at last surgery*** R = 0 R > 0 NA	75/136 22/136 39/136	55.1% 16.2% 28.7%
** *Median Ki67 at diagnosis* **	45.71 (range 1.0–99.9)	
** *No. of patient ongoing at data-cut off ** **	31/136	22.8%
***Vital status at******data-cut off **** Alive Dead	84/136 52/136	61.8% 38.2%
***PARPi*** Niraparib Rucaparib	121/136 15/136	89.0% 11.0%
** *Median age at PARPi start (range)* **	67.3 (range 44.2–86.6)	

* Data cut-off: 30/APR/2022. NA: not available; IDS: interval debulking surgery; ULN: upper limit of the normal range; R: residual disease; PFI: platinum-free interval; PD: progressive disease; CR: complete response; PR: partial response; SD: stable disease.

**Table 2 cancers-15-01032-t002:** PARPi clinical outcomes.

	Niraparib	Rucaparib
**No. of patients**	121/136	15/136
**Starting dose (mg/die)**	300 mg (38/121, 31.4%) 200 mg (83/121, 68.6%)	1200 mg (15/15, 100%)
**Dose reduction** Yes No	66/136 (48.5%) 70/136 (51.5)	
**No. of patients ongoing** **at data cut-off ***	24/32 (77.4%)	7/32 (22.6%)
**Cause of discontinuation** PD Toxicity Other	99/136 (72.8%) 5/136 (3.7%) 1/136 (0.7)	
**Best response to PARPi** RC RP SD PD NA	42/136 (30.9%) 31/136 (22.8%) 29/136 (21.3%) 33/136 (24.3%) 1/136 (0.7%)	

* Data cut-off: 30/APR/2022. NA: not available; PFI: platinum-free interval; PD: progressive disease; CR: complete response; PR: partial response; SD: stable disease.

**Table 3 cancers-15-01032-t003:** Characteristics of the patients according to the median Ki67 value.

Characteristic	Ki67 ≤ 45.7%	Ki67 > 45.7%
** *Median age at diagnosis (range)* **	65.5 (range 38–81)	61 (range 42–84)
** *Median age at PARPi start (range)* **	68.6 (range 44.6–83.6)	64.6 (range 44.2–86.6)
** *International FIGO stage at diagnosis* ** *I* *II* *III* *IV*	3/68 (4.4%) 5/68 (7.4%) 48/68 (70.6%) 12/68 (17.6%)	***p*-value 0.413**
** *Grading* ** *G2* *G3*	1/68 (1.5%) 67/68 (98.5%)	***p*-value 0.559** 2/68 (2.9%) 66/68 (97.8%)
** *CA-125 level at diagnosis* ** *≤* *ULN* *>ULN* *NA*	5/68 (7.4%) 55/68 (80.9%) 8/68 (11.8%)	***p*-value 0.355** 10/68 (14.7%) 49/68 (72.1) 9/68 (13.2%)
** *Type of surgery at diagnosis* ** *Upfront* *IDS* *No surgery*	42/68 (61.8%) 24/68 (35.3%) 2/68 (2.9%)	***p*-value 0.251** 48/68 (70.6%) 20/68 (29.4%) 0 (0%)
** *Residual disease at first surgery* ** *R = 0* *R > 0* *NA*	46/68 (67.6%) 22/68 (32.4%) 0 (0%)	***p*-value 0.405** 50/68 (73.5%) 17/68 (25%) 1/68 (0.5%)
** *CA-125 before starting PARPi* ** *≤* *ULN* *>ULN* *NA*	45/68 (66.2%) 21/68 (30.9%) 2/68 (2.9%)	***p*-value 0.346** 48/68 (70.6%) 20/68 (29.4%) 0 (0%)
** *Histologic type* ** *Serous* *Endometrioid* *Mixed serous and endometrioid*	65/68 (95.6%) 3/68 (4.4%) 0 (0%)	***p*-value 0.269** 61/68 (89.7%) 5/68 (7.4%) 2/68 (2.9%)
** *BRCA wild* ** ** *-* ** ** *type (germline* ** ** *)* ** *Yes* *NA*	62/68 (91.2%) 6/68 (8.8%)	***p*-value 0.195** 57/68 (83.8%) 11/68 (16.2%)
** *BRCA wild* ** ** *-* ** ** *type (somatic* ** ** *)* ** *Yes* *NA*	44/68 (64.7%) 24/68 (35.3%)	***p*-value 0.480** 40/68 (58.8%) 28/68 (41.2%)
** *First PFI* ** *≥12 months* *6–12 months* *NA*	50/68 (73.5%) 17/68 (25.0%) 1/68 (1.5%)	***p*-value 0.828** 53/68 (77.9%) 14/68 (20.6%) 1/68 (1.5%)
** *No. of lines of chemotherapy before* ** * **PARPi** * *2 lines (first recurrence)* *>2 lines* *3 lines* *4 lines* *5 lines* *7 lines*	53/68 (77.9%) 15/68 (22%) 11/68 (16.2%) 2/68 (2.9%) 2/68 (2.9%) 0 (0%)	***p*-value 0.535** 45/68 (66.2%) 23/68 (33.9%) 15/68 (22.1%) 4/68 (5.9%) 3/68 (4.4%) 1/68 (1.5%)
** *Bevacizumab* ** *No* *Yes*	19/68 (27.9%) 49/68 (72%)	***p*-value 0.01** 25/68 (36.8%) 43/68 (63.2%)
** *Type of chemotherapy before PARPi* ** *Carboplatin-liposomial doxorubicine* *Carboplatin-gemcitabine* *Carboplatin monotherapy* *Carboplatin-paclitaxel* *Other*	24/68 (35.3%) 13/68 (19.1%) 0 (0%) 23/68 (33.8%) 8/68 (11.8%)	***p*-value 0.693** 23/68 (33.8%) 10/68 (14.7%) 1/68 (1.5%) 22/68 (32.4%) 12/68 (17.6%)
** *Clinical response after platinum-based chemotherapy before PARPi* ** *CR* *PR* *SD*	18/68 (26.5%) 47/68 (69.1%) 3/68 (4.4%)	***p*-value 0.186** 27/68 (39.7%) 40/68 (58.8%) 1/68 (1.5%)
** *No. of surgeries of before PARPi* ** *0* *1* *2* *3* *5*	1/68 (1.5%) 47/68 (69.1%) 17/68 (25.0%) 8/68 (11.8%) 0 (0%)	***p*-value 0.496** 0 (0%) 42/68 (61.8%) 23/68 (33.8%) 2/68 (2.9%) 1/68 (1.5%)
** *Residual disease at last surgery* ** *R = 0* *R > 0* *NA*	34/68 (50%) 16/68 (23.5%) 18/68 (26.5%)	***p*-value 0.06** 41/68 (60.3%) 6/68 (8.8%) 21/68 (30.9%)
** *Type of chemotherapy after PARPi* ** *Platinum-based chemotherapy* *Liposomial doxorubicin-trabectedin* *Gemcitabine monotherapy* *Liposomial doxorubicin monotherapy* *Paclitaxel monotherapy* *Cyclophosphamide* *Etoposide* *Radiation* *Best supportive care* *Follow-up* *NA*	17/68 (27.9%) 10/68 (16.4%) 3/68 (4.9%) 3/68 (4.9%) 8/68 (13.1%) 0 (0%) 0 (0%) 0 (0%) 2/68 (3.3%) 1/68 (1.6%) 2/68 (3.3%)	***p*-value 0.272** 27/68 (39.7%) 7/68 (10.3%) 1/68 (1.5%) 6/68 (8.8%) 3/68 (4.4%) 2/68 (2.9%) 3/68 (4.4%) 1/68 (1.5%) 3/68 (4.4%) 0 (0%) 3/68 (4.4%)
** *Vital status at data-cut off ** ** *Alive* *Dead*	38/68 (55.9%) 30/68 (44.1%)	***p*-value 0.158** 46/68 (67.6%) 22/68 (32.4%)
** *PARPi* ** *Niraparib* *Rucaparib*	64/68 57/68	***p*-value 0.055** 4/68 11/68

* Data cut-off: 30/APR/2022. NA: not available; IDS: interval debulking surgery; ULN: upper limit of the normal range; R: residual disease; PFI: platinum-free interval; PD: progression disease; CR: complete response; PR: partial response; SD: stable disease.

**Table 4 cancers-15-01032-t004:** PARPi description according to the Ki67 median value.

	Ki67 ≤ 45.7%	Ki67 > 45.7%
**Starting dose (mg/die)** 300 mg 200 mg 1200 mg	17/68 (27.4%) 41/68 (66.1%) 4/68 (6.5%)	***p*-value 0.200** 38/68 (56.7%) 18/68 (26.9%) 11/68 (16.4%)
**Dose reduction** Yes No	33/68 (48,5%) 35/68 (51.5%)	***p*-value 0.200** 33/68 (48.5%) 35/68 (51.5%)
**Cause of discontinuation** PD Toxicity Other Ongoing	45/68 (66.2%) 2/68 (2.9%) 0 (0%) 21/68 (30.9%)	***p*-value 0.115** 54/68 (79.4%) 3/68 (4.4%) 1/68 (1.5%) 10/68 (14.7%)
**Best responce to PARPi** CR PR SD PD NA	13/68 (19.1%) 13/68 (26.5%) 18/68 (26.5%) 19/68 (27.9%) 1/68 (1.5)	***p*-value 0.295** 25/68 (36.8%) 18/68 (26.5%) 17/68 (25%) 11/68 (16.2%) 0 (0%)

PD: progressive disease; CR: complete response; PR: partial response; SD: stable disease; NA: not applicable.

## Data Availability

The data presented in this study are available in the current manuscript “Ki67 as a Predictor of Response to PARP Inhibitors in Platinum Sensitive BRCA Wild Type Ovarian Cancer: The MITO 37 Retrospective Study”.

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
