# Peer review of "Ki67 as a Predictor of Response to PARP Inhibitors in Platinum Sensitive BRCA Wild Type Ovarian Cancer: The MITO 37 Retrospective Study"

_cancers, 2023, doi:10.3390/cancers15041032_

Round 1
Reviewer 1 Report
MITO37 study sought to identify a predictive biomarker able to discriminate which patient with BRCA wild type ovarian cancer would respond better to PARPi. This topic is of great interest to the medical oncology community. However, dr Tuninetti and collegues have failed to demonstrate a role for Ki67 in this field. I think this work can be more interesting after some minor changes:
- introduction: because the authors analyzed niraparib and rucaparib as “PARPi treatment”, I would add some efficacy data that justify this choice (i.e. similar HR in WT BRCA patients?)
- methods: was Ki67 evaluated locally of centrally? Please specify.
- discussion: If Ki67 was analyzed locally, the main limitation of Ki67 is its poor reproducibility for the different antibodies and platforms used. Please discuss it.
Author Response
MITO37 study sought to identify a predictive biomarker able to discriminate which patient with BRCA wild type ovarian cancer would respond better to PARPi. This topic is of great interest to the medical oncology community. However, dr Tuninetti and collegues have failed to demonstrate a role for Ki67 in this field. I think this work can be more interesting after some minor changes:
- introduction: because the authors analyzed niraparib and rucaparib as “PARPi treatment”, I would add some efficacy data that justify this choice (i.e. similar HR in WT BRCA patients?)
Thank you for this suggestion.
The choice of the treatment depends mainly on safety profile since different trials show similar acitivity of the two drugs. Moreover, in Italy, endometrioid ovarian cancers can only be treated with rucaparib. We have added the reference of ARIEL3 and NOVA trial. No data are available on comparison between the two drugs
- methods: was Ki67 evaluated locally of centrally? Please specify.
Ki67 was evaluated locally and this is more clearly specified in the text. (See abstract and Material and Methodsà Ki67 staining and scoring)
- discussion: If Ki67 was analyzed locally, the main limitation of Ki67 is its poor reproducibility for the different antibodies and platforms used. Please discuss it.
Thank you for your comment.
To overcome the possible bias of poor reproducibility, a kick off meeting with the pathologists was performed in order to define the criteria for ki67 evaluation. In the material and methods section it is specified how ki67 was assessed in all pathology units. We added this information in the material and methods section.
Reviewer 2 Report
This is an interesting paper by the MITO group evaluating the predictive role of Ki67 in BRCA1/2wt (?) PSOC patients receiving treatment with PARPi.
The study has an interesting medical hypothesis that we need easily accessible biomarkers to guide treatment with PARPi in this setting. Despite being a negative study, it is an informative one and it is well organized and presented, however there are some issues that need further improvement:
- Judging from the numbers, there is no information regarding the BRCA1/2 status in the tumour for some gBRCA1/2 wt patients. Therefore, authors should comment on this and state that their analysis is limited to gBRCA1/2 wt patients. We cannot exclude that some sBRCA1/2mut pts were included in the analysis
- It is not clearly stated if there was central or local assessment of Ki67. In the later case, authors should comment on possible interlaboratory variability as known from other neoplasms (eg. breast) and its effect on the results of the study.
- Authors have chosen the median as a cutoff value for Ki67 and they are correct that no specific cut off values exist. I would anticipate to see a figure with the ki67 values distribution in the population. In addition, I believe that more detailed analysis is required to understand the role of ki67 eg. evaluating PFS in ki67 quartiles.
- There are formating mistakes in the tables and their abbreviations that should be corrected
- The current analysis evaluated the predictive and not the prognostic role of Ki67 and this should be stated across the test. Phrases including the prognostic significance of Ki67 should be deleted
- No patients received Olaparib, authors should comment on that.
